# Distinct Predictors of Clinical Response after Repetitive Transcranial Magnetic Stimulation between Bipolar and Unipolar Disorders

**DOI:** 10.3390/ijerph20075276

**Published:** 2023-03-27

**Authors:** Aurélie Lacroix, Aude Paquet, Mireille Okassa, Théodore Vinais, Marilyne Lannaud, Brigitte Plansont, Alexandre Buisson, Sandrine Guignandon, Dominique Malauzat, Murielle Girard, Benjamin Calvet

**Affiliations:** 1Unité de Recherche et d’Innovation, Centre Hospitalier Esquirol, 87025 Limoges, France; 2INSERM, U1094 Institut d’Epidémiologie et de Neurologie Tropicale, Université de Limoges, CHU Limoges, IRD U270, GEIST, 87000 Limoges, France; 3Centre de Recherche en Epidémiologie et Santé des Populations, U1018 INSERM, Paris-Saclay Université, UVSQ, 94800 Villejuif, France

**Keywords:** rTMS, bipolar disorder, unipolar disorder, drug-resistant depression, care, clinical response, predictors

## Abstract

*Background:* Repetitive transcranial magnetic stimulation (rTMS) has been shown to be therapeutically effective for patients suffering from drug-resistant depression. The distinction between bipolar and unipolar disorders would be of great interests to better adapt their respective treatments. *Methods:* We aimed to identify the factors predicting clinical improvement at one month (M1) after the start of rTMS treatment for each diagnosis, which was preceded by a comparison of the patients’ clinical conditions. We used the data collected and the method employed in a previous publication on 291 patients. *Results:* Although the bipolar group had fewer responders, these patients seemed to better maintain their post-rTMS improvement on anxiety and perception of the severity of their illness than those in the unipolar group. For the bipolar group, young age coupled with low number of medications and high fatigue was shown to be the best combination for predicting improvement at M1. The duration of current depressive episode, which was previously demonstrated for whole group, combined with being attached was shown to favor clinical improvement among the patients in unipolar group. *Conclusion:* We were able to define a combination of specific factors related to each diagnosis for predicting the patients’ clinical response. This could be extremely useful to predict the efficacy of rTMS during routine clinical practice in neuromodulation services.

## 1. Introduction

Repetitive transcranial magnetic stimulation (rTMS) has been shown to be therapeutically effective in a variety of clinical disorders, including those associated with drug-resistant depressive episodes, with a sufficient level of evidence [1,2,3,4]. Factors predicting clinical response in these patients are frequently associated with the overall depressive population or only with the unipolar population, which constitutes the majority of the samples [5,6]. Indeed, unipolar and bipolar disorders are often mixed up in the literature, even though they are clinically distinct entities. Thus, it would be crucial to distinguish them to better predict the adaptation of patients to potential treatment. Besides their medical diagnosis, the distinction between unipolar and bipolar disorders is not well defined since the bipolar population remains relatively small but is also very complex in its classification (Bipolar type 1, Bipolar type 2, Cyclothymic disorder, Mixed episodes, Rapid cycling, and Psychotic features), often consisting of very few individuals in very large cohorts [7]. Considering the growing interest in the use of rTMS for the treatment of bipolar disorder [8], we believe that it would be crucial to distinguish populations that are treated in identical ways with standardized instruments but have clinically different diagnoses. In this study, we focused on the potential factors predicting clinical response to rTMS to add to the limited literature, particularly with respect to the distinction between bipolar and unipolar disorders.

We conducted a new analysis comparing patients with bipolar and unipolar disorders based on a retrospective naturalistic study performed in the Neuromodulation Department of the Esquirol Hospital in Limoges. This study included patients with drug-resistant depression who were selected from an active file of more than 600 patients referred for rTMS for various pathologies since 2007 [9]. We compared these two distinct types of disorders using the same methodology employed in the original study, which included the Clinical Global Response (CGI), Beck Depression Inventory (BDI), Hamilton Rating Scale for Depression (HDRS), and Perceived Deficits Questionnaire scores. Clinical response was defined as a decrease in the HDRS score of at least 50% from inclusion (D0) to one month (M1). Patients with bipolar and unipolar disorders were examined separately to identify the factors predicting clinical response at M1 after the beginning of rTMS.

## 2. Methods

### 2.1. Population

Patients over 18 years of age with a diagnosis of uni- or bipolar disorder by a psychiatrist, according to the DSM-IV-TR and DSM-5 criteria, since 2007 were included in the study. Specifically, patients with a bipolar affective disorder (F31 ICD-10) versus a unipolar disorder, such as depressive episodes (F32 ICD-10), recurrent depressive disorder (F33 ICD-10), or persistent [affective] mood disorders (F34 ICD-10), were included. To be eligible, their depressive episode had to be characterized by a HDRS score ≥ 8 at D0 and correspond to drug-resistant depression (at least minor and/or major), i.e., the failure of at least two drug treatments (antidepressants, anxiolytics, hypnotics, antipsychotics, and mood stabilizers) prior to their treatment with rTMS. More specifically, a minimum HDRS score of 16 at D0 is generally considered to be the reference to indicate moderate depression [10]. Thus, we distinguished between patients with the so-called mild depression, with an HDRS score between 8 and 15, and those with major depression, with an HDRS score ≥ 16. Finally, their inclusion in the study required that they completed a course of rTMS treatment (low frequency, with right dorsolateral prefrontal cortex stimulation) comprising five sessions per week for three weeks, along with complete data on the relevant psychometric scales. The only contraindications were those classically associated with rTMS treatment: the presence of a pacemaker, a cardiac or cerebral stimulator, or metal splinters in the head; brain injury; active comitiality; pregnancy; and intracranial hypertension. Participation in another biomedical research study was also not allowed. The clinical evaluations and corresponding data have been collected since 2007 in the patients’ files. Permission to use the participants’ research data was obtained retrospectively for those included before 2018 and prospectively since then. All patients provided informed written consent, and the study received legal authorizations from the Committee for the Protection of Persons and the French Agency for the Security of Health Products.

### 2.2. rTMS Treatment

The treatment included five daily sessions per week for three weeks. A daily session consisted of 20 min of stimulation at a low frequency of 1 Hz (Magstim Super Rapid^2^, Inomed, Emmendingen, Germany), with 120 trains of 10 s of stimulation (60 pulses) and an inter-train interval of 1 s. This corresponded to 1200 stimulations per session at 100% of the resting motor threshold (RMT). This threshold corresponded to the intensity of the magnetic field that was necessary and sufficient to produce 5 contralateral motor-evoked potentials of at least 50 microvolts in amplitude out of 10 magnetic stimuli of the region concerned. The cerebral area that was stimulated was the right dorsolateral prefrontal cortex. A neuronavigation process (Visor 3D, ANT, Enschede, The Netherlands) allowed real-time monitoring of the specific position of stimulation based on medical imaging obtained by magnetic resonance imaging (MRI) (Asalab, ANT, Enschede, The Netherlands). An overlay corresponding to a standard MRI (resulting from the averaging of the MRI of different subjects using Collins brain) with the positional coordinates of the coil and the head in real time was performed. The area to be stimulated could then be marked and retrieved at each session.

### 2.3. Psychometric Evaluations

The CGI [11] is a global evaluation that uses three independent item scales. It is easy to use and generalizable to all pathologies or co-morbidities. The scales consist of the severity of illness (0 to 7), the global clinical response after treatment (0 to 7), and a composite score on a four-point scale that addresses treatment efficiency and adverse effects.

The score of item 1 of the CGI at D0 and the scores of items 1 and 2 at M1, which were taken separately, were analyzed.

The BDI is a 21-item, self-reported rating inventory that measures the intensity of depression [12]. Each item is comprised of four sentences corresponding to four increasing degrees of intensity with regard to a symptom and is scored from 0 to 3. The highest score obtained is the score selected for the same set. The global score is determined by summing the scores of the 21 items. A score < 10 indicates the absence of depression; a score from 10 to 18 indicates mild depression; a score from 19 to 29 indicates moderate depression; and a score >30 indicates severe depression.

The overall BDI scores, as well as the scores for items L (Fatigue) and M (Appetite), at D0 and M1 were analyzed.

The HDRS [13,14] is a 17-item instrument that was designed to measure the frequency and intensity of depressive symptoms in individuals with major depressive disorder and is completed by a clinician. The ratings are made using a Likert scale ranging either from 0 to 4 or from 0 to 2 for each item, yielding total scores from 0 to 52. HDRS scores are classified as normal (0 to 7), mild depression (8 to 16), mild to moderate depression (17 to 23), and moderate to severe depression (>24).

The overall HDRS global score, the sleep (items 4 + 5 + 6) and anxiety (items 10 + 11) sub-scores, and the depressive core (HDRS_6_: 1 + 2 + 7 + 8 + 10 + 13), in addition to the scores for items 8 (slowdown), 9 (agitation), 13 (general symptoms), 15 (hypochondria), and 16 (weight loss), at D0 and M1 were analyzed.

The questionnaire on perceived deficits [15] includes five items that can be scored from 0 to 4 for each item (0 corresponds to not once and 4 corresponds to very often). This questionnaire concerns memory, attention, or concentration problems that some people may have. These situations are assessed based on a maximum score of 20 in the last seven days.

### 2.4. Clinical Response

The gold standard definition for clinical response, which is consistent with the data in the literature [9], consists of a calculated clinical response of at least a 50% reduction in the HDRS score between the assessment at inclusion and the assessment at M1.

### 2.5. Statistical Analysis

The quantitative variables are presented as means and standard deviations. The qualitative variables are presented as percentages and counts. The inter-group comparisons for the quantitative variables were performed using the Mann–Whitney non-parametric tests for a given time point. The chi-squared test was used to compare the groups for the qualitative variables. A binary stepwise logistic regression model was generated to identify the predictive factor(s) (variables collected at D0 as a predictive factor of the clinical response at M1). Variables that differed between clinical response groups with *p* < 0.2 were introduced into the binary logistic regression model. Results with *p*-values < 0.05 were considered significant. The analyses were performed using the SPSS Statistics 27.0 software (IBM, Bois-Colombes, France).

## 3. Results

### 3.1. Comparison between Bipolar and Unipolar Disorders

Of the 291 patients included in the study, most of the diagnoses corresponded to a unipolar disorder as represented by F32 ICD-10, F33 ICD-10, and F34 ICD-10 (86.6%, n = 252), and very few diagnoses corresponded to a bipolar disorder as represented by F31 ICD-10 (13.4%, n = 39).

#### 3.1.1. At Inclusion (Table 1)

The BDI global score (20.05 ± 0.400 vs. 17.05 ± 1.022, *p* = 0.013), HDRS global score (17.75 ± 0.272 vs. 15.49 ± 0.671, *p* = 0.003), HDRS sleep sub-score (items 4 + 5 + 6) (2.33 ± 0.087 vs. 1.77 ± 0.245, *p* = 0.016), and scores for HDRS items 9 (agitation) (0.39 ± 0.045 vs. 0.18 ± 0.089, *p* = 0.038) and 15 (hypochondria) (1.00 ± 0.060 vs. 0.69 ± 0.128, *p* = 0.046) were statically higher in patients with a unipolar disorder than in patients with a bipolar disorder. The number of previous medications was statically lower in the patients with a unipolar disorder than in the patients with a bipolar disorder (4.61 ± 0.161 vs. 5.18 ± 0.326, *p* = 0.025) (Table 1).

**Table 1 ijerph-20-05276-t001:** Clinical data and psychometric scales scores comparison between unipolar and bipolar at D0 and at M1.

*Variables*	D0			M1		
	Unipolar	Bipolar	*p*	Unipolar	Bipolar	*p*
*Age (Years mean ± SD)*	55.37 ± 0.809	53.67 ± 1.753	0.491			
Sex ratio (*men number/women number*)	0.51	0.44	0.715			
Attached (*n* (%))	138 (54.8)	21 (53.8)	0.854			
Children (*n* (%))	83 (32.9)	16 (41)	0.289			
Work (*n* (%))	140 (55.6)	21 (53.8)	0.639			
ECT (*n* (%))	23 (9.1)	4 (10.3)	0.769			
Antidepressant previous treatment (*Number mean ± SD*)	3.05 ± 0.136	3.05 ± 0.136	0.986			
Total previous treatment (*Number mean ± SD*)	4.61 ± 0.161	5.18 ± 0.326	0.070			
Current depressive episode (*Months mean ± SD*)	7.24 ± 1.022	7.10 ± 1.299	0.599			
HDRS Total (*Score mean ± SD*)	17.75 ± 0.272	15.49 ± 0.671	**0.003**	9.75 ± 0.367	9.36 ± 0.816	0.851
HDRS Sleep (*Score mean ± SD*)	2.33 ± 0.087	1.77 ± 0.245	**0.016**	1.17 ± 0.079	1.26 ± 0.234	0.967
HDRS Anxiety (*Score mean ± SD*)	2.62 ± 0.090	2.31 ± 0.239	0.279	1.62 ± 0.090	1.08 ± 0.199	**0.010**
HDRS Depressive core (*Score mean ± SD*)	8.98 ± 0.179	8.51 ± 0.491	0.440	4.97 ± 0.218	4.77 ± 0.485	0.982
HDRS item 8 (*Score mean ± SD*)	1.09 ± 0.053	1.10 ± 0.141	0.996	0.49 ± 0.041	0.67 ± 0.148	0.544
HDRS item 9 (*Score mean ± SD*)	0.39 ± 0.045	0.18 ± 0.089	**0.038**	0.31 ± 0.040	0.38 ± 0.108	0.362
HDRS item 13 (*Score mean ± SD*)	0.96 ± 0.044	1.08 ± 0.124	0.334	0.58 ± 0.045	0.62 ± 0.108	0.628
HDRS item 15 (*Score mean ± SD*)	1.00 ± 0.060	0.69 ± 0.128	**0.046**	0.75 ± 0.053	0.85 ± 0.101	0.174
HDRS item 16 (*Score mean ± SD*)	0.50 ± 0.040	0.46 ± 0.109	0.529	0.15 ± 0.026	0.13 ± 0.066	0.725
BDI Total (*Score mean ± SD*)	20.05 ± 0.400	17.05 ± 1.022	**0.013**	9.94 ± 0.424	9.53 ± 1.159	0.690
BDI item L (*Score mean ± SD*)	1.77 ± 0.048	1.71 ± 0.130	0.623	0.99 ± 0.047	0.94 ± 0.112	0.716
BDI item M (*Score mean ± SD*)	1.04 ± 0.060	0.79 ± 0.147	0.115	0.45 ± 0.047	0.28 ± 0.094	0.178
CGI 1 (*Score mean ± SD*)	5.34 ± 0.036	5.26 ± 0.071	0.277	4.33 ± 0.063	3.97 ± 0.174	**0.046**
CGI 2 (*Score mean ± SD*)				3.44 ± 0.066	3.23 ± 0.162	0.194
Perceived Deficits Questionnaire (*Score mean ± SD*)	9.86 ± 0.920	11.00 ± 1.528	0.644	7.53 ± 0.850	9.33 ± 0.882	0.542

#### 3.1.2. At M1 

The HDRS anxiety sub-score (items 10+11) (1.62 ± 0.090 vs. 1.08 ± 0.199) was statically higher in the patients with a unipolar disorder than in the patients with a bipolar disorder (*p* = 0.010). The severity of the illness corresponding to CGI item 1 (4.33 ± 0.063 vs. 3.97 ± 0.174) was also statically higher in the patients with a unipolar disorder than in the patients with a bipolar disorder (*p* = 0.046) (Table 1).

#### 3.1.3. Clinical Response: Gold Standard with 50% Decrease in HDRS at M1

A clinical response with the gold standard decrease of 50% in the HDRS was observed at M1 for 46.4% of the unipolar population (n = 117), who were considered to be the responders, whereas this clinical response was only observed for 28.2% of the bipolar population (n = 11). The observed difference between the two diagnoses was significant (*p* = 0.033).

### 3.2. Predictive Factors at M1 

Among the variables (age, number of antidepressant treatments, number of drugs, BDI at D0 (total, item L (Fatigue) and M (Appetite)), HDRS at D0 (total, sleep and anxiety sub-scores, depressive core (HDRS_6_: 1 + 2 + 7 + 8 + 10 + 13), items 8 (slowdown)-9-13 (general symptoms)-15-16 (weight loss)), CGI item 1 at D0, number of previous courses of rTMS, duration of the current depressive episode, perceived deficits at D0, sleepiness, diagnosis, professional activity, being attached, having children, ECT prior to the rTMS treatment, and sex), those that were not correlated with *p* < 0.2 based on the clinical response criteria at M1 were introduced into a regression model. The analysis of clinical response only employed the gold standard definition (Table 2).

#### 3.2.1. For Bipolar Disorder Only 

The qualified variables included the BDI item L score at D0 (group without a clinical response: 1.56 ± 0.185 vs. group with a clinical response: 2.25 ± 0.250, *p* = 0.025), the total score on the HDRS at D0 (group without a clinical response: 15.00 ± 1.038 vs. group with a clinical response: 17.88 ± 1.074, *p* = 0.078), the duration of the current depressive episode (group without a clinical response: 8.67 ± 1.696 months vs. group with clinical response: 3.56 ± 1.132 months, *p* = 0.054), the number of drugs (group without a clinical response: 6.17 ± 0.493 vs. group with a clinical response: 4.25 ± 0.675, *p* = 0.026), having children (group without a clinical response: 0.33 ± 0.114 vs. group with a clinical response: 0.75 ± 0.164, *p* = 0.028), and age (group without a clinical response: 56.17 ± 2.952 years vs. group with a clinical response: 49.38 ± 3.525 years, *p* = 0.040). After the introduction of these variables into a stepwise binary logistic regression, the number of drugs, alone or in combination with the BDI item L score at D0, or these two variables in combination with the age of the patients, were predictive of the patients’ clinical response at M1 for the patients suffering from bipolar disorder.

However, the number of drugs combined with the BDI item L score at D0 and the age of the patients at inclusion explained 68.9% of the variance in clinical response at M1. The regression coefficients associated with the number of drugs (Exp(B) = 0.155, ß = −1.863, *p* = 0.041) and the age of the patients (Exp(B) = 0.846, ß = −0.167, *p* = 0.107) were negative, whereas the regression coefficient associated with the BDI item L score (Exp(B) = 13.377, ß = 2.594, *p* = 0.063) was positive. Thus, a lower number of drugs used, combined with a lower age of the patients but a higher score on fatigue for the BDI item L (*p* < 0.001), was the most discriminating combination of factors predictive of the patients’ clinical response at M1 among those observed (Table 2).

#### 3.2.2. For Unipolar Disorder Only 

The qualified variables included the number of antidepressant medication (group without a clinical response: 2.84 ± 0.236 vs. group with clinical response: 2.93 ± 0.225, *p* = 0.107), the BDI item L score at D0 (group without a clinical response: 1.92 ± 0.084 vs. group with clinical response: 1.70 ± 0.098, *p* = 0.065), the total score on the HDRS at D0 (group without a clinical response: 17.88 ± 0.520 vs. group with a clinical response: 18.20 ± 0.561, *p* = 0.137), the number of previous courses of rTMS (group without a clinical response: 1.13 ± 0.047 vs. group with clinical response: 1.25 ± 0.078, *p* = 0.118), the duration of the current depressive episode (group without a clinical response: 8.96 ± 1.687 months vs. group with clinical response: 4.92 ± 0.744 months, *p* = 0.015), being attached (group without a clinical response: 0.57 ± 0.057 vs. group with a clinical response: 0.39 ± 0.066, *p* = 0.114), and sleepiness (group without a clinical response: 0.13 ± 0.039 vs. group with a clinical response: 0.04 ± 0.025, *p* = 0.035). After the introduction of these variables into a stepwise binary logistic regression, being attached, alone or in combination with the duration of the current depressive episode, was predictive of the patients’ clinical response at M1.

However, being attached and the duration of the current depressive episode explained 10.6% of the variance in clinical response at M1. The regression coefficients associated with being attached (Exp(B) = 2.171, ß = 0.775, *p* = 0.034) was significantly positive, whereas the regression coefficient associated with the duration of the current depressive episode (Exp(B) = 0.930, ß = −0.072, *p* = 0.040) was significantly negative. Thus, being more attached combined with a lower duration of the current depressive episode (*p* = 0.004) was the most discriminating combination of factors predictive of the patients’ clinical response at M1 among those observed (Table 2).

## 4. Discussion

We initially studied patients with unipolar and bipolar disorders through their common diagnosis of drug-resistant depression, as published in a previous publication. Subsequently, a comparative evaluation of the two pathologies was performed, considering their clinical individuality and different responses to rTMS. This evaluation aimed to elucidate predictive factors of clinical response to rTMS.

To date, only a few studies have examined these aspects. Indeed, the variables examined our study can be found in the literature individually, but not collectively.

In our naturalistic study, the differences at inclusion showed that although bipolar patients consumed significantly more medications before the start of rTMS, they seemed to suffer less from depression when compared to unipolar patients based on the BDI and HDRS scores (statistically significant differences). In bipolar diagnosis, in addition to rTMS’s mood stabilization role [16], similar findings regarding BDI [17] and HDRS [18] scores have been demonstrated in the literature. However, these findings still remain controversial.

As in our research, a study by Sung et al. [18] distinguished between the sub-scores and items of the HDRS. They observed that the sub-score related to sleep and insomnia was significantly higher in unipolar patients than in bipolar patients. Similar results were observed for item 15, indicating that the unipolar group was more hypochondriac than the bipolar group. Concerning item 9, the agitation level was significantly higher in the unipolar group than in the bipolar group. Similar findings have not been reported in the literature for item 9, but a parallel can be drawn with the sleep sub-score and score on item 15, where the insomniac and hypochondriac side of bipolar disorder can be correlated with a certain amount of stress that can induce agitation in the patients.

Recent publications have demonstrated no difference in the HDRS score between the groups [19], but they underline an interest in studying the effectiveness of rTMS in a distinctive manner through these two diagnoses [8].

Alhelali et al. demonstrated a difference in the effect of rTMS between bipolar and unipolar patients with respect to items 10 and 11 [8]. Similarly, we observed a difference in the effect of rTMS with respect to the anxiety sub-score while grouping the items together. This sub-score was significantly lower in the patients with a bipolar disorder at M1 after the rTMS intervention, which seems to be an excellent marker of the effectiveness of rTMS in this group. The absence of this difference at inclusion reinforces the use of this marker to compare the effectiveness of rTMS between the groups. Another marker representing the patients’ own feelings concerning the severity of the illness can be added in the presence of item 1 of the CGI, which was found to be significantly lower in the bipolar group. This marker has not been used previously to distinguish the effectiveness of rTMS between bipolar and unipolar patients. However, it seems to be a marker of interest, especially in association with the hetero-questionnaire of lower anxiety. These two factors (anxiety perceived by the patients and their subjective evaluation of the severity of the illness) were found to be significantly lower at M1, suggesting the maintenance effect of rTMS in patients with bipolar disorder.

These findings are not consistent with the effectiveness of rTMS in the bipolar group, which had significantly fewer responders at M1 than the unipolar group. Thus, it can be concluded that despite the low number of responders, the bipolar group exhibited a better quality of rTMS maintenance, particularly in terms of their anxiety and feelings concerning the severity of their illness.

Thus, the predictive factors of the effectiveness of rTMS in these two pathologies appears to be of notable interest.

In the bipolar group, young age coupled with a low number of medications and high patient fatigue predicted a large variance of 70% in terms of clinical improvement at M1 after the rTMS treatment. However, this finding is unique to bipolar disorder; since this combination of factors has never been studied, each of these factors may have been discussed individually in the literature. For example, an association between age and clinical improvement was found in a study [20] involving 56 patients with bipolar disorder. Young age was found to be a predictive factor for the success of rTMS in a retrospective clinical trial [21] involving a mixed population of bipolar and unipolar patients, which is consistent with our results. Indeed, in previous studies, the bipolar population was often mixed with other patients suffering from depression, and the predictive factors uniquely associated with bipolar disorder were not studied.

The notion of great fatigue has also been found to be associated with clinical improvement in several studies [22,23], wherein the axis of sleep disorder and retardation were investigated as predictive factors for good clinical response to rTMS. However, as previously mentioned, these studies included mixed populations of unipolar and bipolar patients and contained only eight and four patients with bipolar disorder, respectively.

In a study by Polezczyk et al. [24], a history of treatment with several medications was found to be a predictive factor for good response to rTMS in 30 patients with bipolar disorder. However, this study included a mixed population of unipolar and bipolar patients.

The unipolar group validates our study results, since we found that a short duration of current depressive episode was a predictive factor for improvement at M1 after the rTMS treatment, a finding consistent with that reported in our previous study [9]. Indeed, the patient population in our initial study was similar to the unipolar population, since the majority of our sample consisted of unipolar patients. In a novel way, being attached, or the security aspect of this status, could influence the prediction of successful rTMS. However, this combination including duration of depressive episode and being attached has never been discussed in previous literature. Caution is advised, since the variance in clinical improvement among the unipolar patients was only 10% due to the presence of this combination. As stated in the comparative analysis, despite the small population, the evidence of factors predicting the variance in clinical improvement seems stronger at nearly 70% in the bipolar population than in the unipolar population.

The aforementioned combinations of factors predicting clinical improvement after rTMS is of great interest, especially since the duration of this study ensured the stability of data over time.

The strong points of this study are the definitions of new combinations of factors predicting the effectiveness of rTMS in neuromodulation services for patients suffering from bipolar disorders and unipolar disorders in the context of drug-resistant depression.

The weak point could be the small number of bipolar patients, which is a limitation that is present in all existing publications. However, the combination of factors predicting the effectiveness of rTMS for bipolar patients has a very important statistical strength, which is more important than that found for unipolar patients even though there are more unipolar patients in our cohort.

## 5. Conclusions

We were able to define a combination of specific factors predicting clinical response to rTMS among patients with bipolar disorder, which is distinct from the factors associated with unipolar disorder. This could be extremely useful in predicting the efficacy of rTMS in routine clinical practice, leading to clearer treatment choices for patients with drug-resistant depression through distinguishing bipolar disorder from unipolar disorder in neuromodulation services. Our recommendations would be to better observe each combination of specific factors predicting clinical response in each of the targeted diagnoses in order to better adapt the non-drug treatment options proposed to these patients. Transcranial direct current stimulation (tDCS) and psychomotor treatments could be solutions to consider in the absence of this combination of specific factors, which could predict clinical improvement after rTMS treatment.

## Figures and Tables

**Table 2 ijerph-20-05276-t002:** Predictive factors (stepwise binary logistic regression) at M1 under gold standard for bipolar disorder only (a) and for unipolar disorder only (b).

**a.**						
	**β**	**ES**	**Wald**	**ddl**	** *p* **	**Exp(B)**
BDI item L D0	2.594	1.395	3.458	1	0.063	13.377
Number of drugs	−1.863	0.912	4.175	1	0.041	0.155
Age	−0.167	0.103	2.596	1	0.107	0.846
* Note * : global % = 76.9, χ^2^ = 17.417, R^2^ Nagelkerke = 0.689, ddl = 3, *p* < 0.001.						
**b.**						
	**β**	**ES**	**Wald**	**ddl**	** *p* **	**Exp(B)**
Duration of the current depressive episode	−0.072	0.035	4.227	1	0.040	0.930
Be attached	0.775	0.367	4.473	1	0.034	2.171
* Note * : global % = 65.4, χ^2^ = 10.912, R^2^ Nagelkerke = 0.106, ddl = 2, *p* = 0.004.						

## Data Availability

The data presented in this study are available from the corresponding author upon request.

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
