# Peer review of "Distinct Predictors of Clinical Response after Repetitive Transcranial Magnetic Stimulation between Bipolar and Unipolar Disorders"

_ijerph, 2023, doi:10.3390/ijerph20075276_

Round 1

Reviewer 1 Report

Would you rephrase the title please? grammar and structure are unclear. 

1 month ... Do you mean (Month 1) or the first month of rTMS? 

in lines 39.40.41 I completely disagree. In psychiatry, the bipolar NOT simply classfied as unipolar and bipolar! It is more complex. Bipolar disorder classified into the following subclasses: Bipolar type 1, Bipolar type 2, Cyclothymic disorder, Mixed episodes, Rapid cycling, and Psychotic features. Therefore, please rewrite your introduction accordingly. 

You need to mention with justifications of why you use these scores for the measuring the depression score? Why NOT you use the bipolar scales? such as (Clinical Global Impression Scale - Biploar version), OR (Bech-Rafaelsen Mania Scale), OR (Mood disorder questionnaire), OR (Bipolar Depression Rating Scale)? 

You mentioned that the distinguish between unipolar and bipolar is unclear.... then how you distinguish and differentiate between them? please mention that in detail.  

Line 54 and 55. Do you mean if the score of the scales you mentioned less than 50% from the initial assessment, then the rTMS is successful? please clarify. 

You need to mention which score indicated and confirm the uni or bi polar disorders in the scales you mentioned in method.

You wrote many times (drug-resistant depression), would you mention which drugs, are they antidepressants or anticonvulsants or antipsychotics. As the recent evidence based medicine stated the combination of olanzapine + fluoxetine is an option for this condition and giving good outcomes.

What the external validity of the participants in regards the diagnosis of unipolar or bipolar. For example, are they diagnosed by their doctors or specialists before they enter into the research? Do you think all the participants have been confirmed with drug-resistant depression? That means they tried more than 3 or 4 options of treatment with no improvement?

In line 203, one of the result stated (group without clinical responses 0.57 +- 0.057) how you interpret this?

What are the weak point and strength point of your research? You need to mention that please

Based on the results and outcomes, what are you recommendations need to do for further research in this field and why?

Reviewer 2 Report

Authors of the manuscript aimed to identify the predictors of treatment response in bipolar and unipolar disorder following the rTMS application for drug-resistant depression. In its current form, the manuscript lacks a clear description of the aim and hypothesis, important methodological details, appropriate statistical and viisual representation of the obtained results.

Major points

1. Introduction. Overall the aim and hypothesis of this study is not clearly stated. Why authors want to distinguish patients with uni- and bipolar disorders based on rTMS improvement if this can be done by using clinical criteria (e.g. DSM-5)?

2. Methods. Line 60. The terminology should be consistent throughout the whole manuscript – bipolar/unipolar disorder or depression

3. Methods. Line 78. Here is mentioned that data were collected since 2017 but in line 61 since 2016. State more clearly the year of data collection.

4. Methods. Line 79. What does it mean that consent to use research data was obtained ‚retrospectively‘?

5. Methods. Figure 1. First, this figure can not be called flow-chart, since it shows only the inclusion criteria, which were outlined in the main text. Hence, it is not relevant as it does not show any additional information. Second, the quality oft he figure is low

6. Methods. rTMS treatment. It is not mentioned how RMT was determined.

7. Methods. rTMS treatment. It is not clear if individual MRI was used during the neuronavigation.

8. Methods. Psychometric evaluations. Why namely scores of L (Fatigue) and M (Appetite) items were analyzed?

9. Results. Important clinical information is missing. Age, gender distribution, disease duration, etc. are missing. These data should be added in Table 1.

10. Results. Predictive factors at M1 (Table 2). Why authors decided to include such variables as professional activity, attached (what does this mean?), and children into the regression model? Are these variables relevant?

11. Results. Predictive factors at M1 (Table 2). With such many factors included into the logistic regression model, there is a high probability of multicollinearity.

12. Results. Predictive factors at M1 (Table 2). ‚those with p < 0.2 based on the clinical response criteria at M1 and not correlated were introduced into a regression model‘. What does this mean? This sentence is not clear.

13. Results. Overall, the outputs of binary regression models are not presented correctly in the main text. The most important statistics to be presented in the text is the Exp(B) or odds ratio with confidence intervals but not the mean values (±SD) and regression coefficients (which can be left in tables).

14. Results. Statistical details on longitudinal differences on psychomotor scales for both groups are missing. Only cross-sectional comparison at both time points between the groups is presented. Did authors perform any Wilcoxon or paired t-tests?

Minor points

1. Introduction. Lines 38 - 39. ‚Thus, it would be interesting to distinguish them to predict the adaptation of potential treatment‘. Why ‚interesting‘? It is ‚important‘ or ‚crucial‘ to distinguish. And what does here ‚adaptation‘ to treatment mean?   

2. Introduction. Line 43. Here the same as above ‚it would be interesting‘. Should be rephrased

3. Methods. Line 78. Change the word ‚non-opposition‘

4. Methods. Statistical analysis. The whole paragraph is in italic. Should be changed

5. Methods. Line 134. ‚were introduced into the regression model‘, do authors mean here into the binary logistic regression model?

6. Results. Line 145. ‚were statically different‘. It should be indicated clearly if these values were higher or lower in either group

7. Results. Line 150. Here the same as in previous point ‚was statically different‘.

Round 2

Reviewer 1 Report

I think it is ready for publication as the authors responses to all my enquiries and corrections. 

Reviewer 2 Report

No comments